# Autologous Platelet-Rich Plasma Administration on the Intervertebral Disc in Low Back Pain Patients with Modic Type 1 Change: Report of Two Cases

**DOI:** 10.3390/medicina59010112

**Published:** 2023-01-05

**Authors:** Soya Kawabata, Kurenai Hachiya, Sota Nagai, Hiroki Takeda, Mohd Zaim Mohd Rashid, Daiki Ikeda, Yusuke Kawano, Shinjiro Kaneko, Yoshiharu Ohno, Nobuyuki Fujita

**Affiliations:** 1Department of Orthopedic Surgery, School of Medicine, Fujita Health University, 1-98 Dengakugakubo, Kutsukake-cho, Toyoake 470-1192, Japan; 2Department of Spine and Spinal Cord Surgery, School of Medicine, Fujita Health University, Toyoake 470-1192, Japan; 3Department of Radiology, School of Medicine, Fujita Health University, Toyoake 470-1192, Japan; 4Joint Research Laboratory of Advanced Medical Imaging, School of Medicine, Fujita Health University, Toyoake 470-1192, Japan

**Keywords:** platelet-rich plasma, Modic type 1, low back pain, regenerative medicine

## Abstract

*Background and Objectives*: Modic type 1 is known to be associated with lower back pain (LBP), but at present, a treatment has not been fully established. Meanwhile, platelet-rich plasma (PRP) has been used for tissue regeneration and repair in the clinical setting. There is no clinical PRP injection trial for the intervertebral disc of LBP patients with Modic type 1. Thus, this study aimed to verify PRP injection safety and efficacy in LBP patients with Modic type 1. As a preliminary experiment, two LBP cases with Modic type 1 are presented. *Materials and Methods*: PRP was administered intradiscally to two LBP patients with Modic type 1. PRP was obtained from the patients’ anticoagulated blood. Primary endpoints were physical condition, laboratory data, and X-ray for safety evaluation. Secondary endpoints were pain scores using the visual analog scale (VAS), the Oswestry Disability Index (ODI), and the Roland–Morris Disability Questionnaire (RDQ) to evaluate PRP efficacy. The observation period was 24 weeks after the PRP injection. In addition, changes in Modic type 1 using MRI were evaluated. *Results*: This study assessed two LBP patients with Modic type 1. There were no adverse events in physical condition, laboratory data, or lumbar X-rays after injection. Follow-up MRI showed a decrease of high signal intensity on T2WI compared to before PRP administration. The pain scores tended to improve after the injection. *Conclusions*: PRP injection into the intervertebral disc of LBP patients with Modic type 1 might be safe and effective. This analysis will be continued as a prospective study to establish the efficacy.

## 1. Introduction

Low back pain (LBP) is a common complaint encountered in general practice and is the most common complaint among symptoms originating from the musculoskeletal system [1]. Based on a World Health Organization estimation, 80% of people > 60 years of age had LBP at some point in their lives [2]. LBP significantly affects medical, economic, and social status. There are numerous reports indicating that LBP leads to long periods of absence from work and makes return to society challenging [3]. In addition, a previous study showed that, on average, patients with chronic LBP have a 7% shorter life expectancy, and healthy participants chose a 10% shorter life expectancy to avoid chronic LBP [4]. Therefore, it is imperative to develop a novel strategy for LBP treatment.

Modic et al. reported intensity changes in the endplates and subchondral bone on MRI as Modic change and was classified into three types [5,6]. Among the three Modic change types, type 1 shows low signal intensity on T1WI and high signal intensity on T2WI, indicating edema and hypervascularity in the vertebral body. Although a significant correlation between Modic type 1 and LBP has been described [7], there has been no established treatment for Modic type 1.

Platelet-rich plasma (PRP) has been used for tissue regeneration and repair in the clinical setting. Recently, especially in the field of orthopedics, PRP has demonstrated a regenerative ability to repair injured tissues, including tendons, ligaments, and cartilage [8,9,10,11]. There is no clinical PRP injection trial for the intervertebral disc of LBP patients with Modic type 1. Thus, this study aimed to verify PRP injection safety and efficacy in LBP patients with Modic type 1. As a preliminary experiment, two LBP cases with Modic type 1 were presented.

## 2. Materials and Methods

### 2.1. Patients

This study includes two patients who met the inclusion and exclusion criteria (Table 1). The consenting patients were followed up 24 weeks from administration.

### 2.2. PRP Preparation and Procedure

To create LR(leukocyte-rich)-PRP, 30 mL of anticoagulated blood from the patient was initially collected (ACD-A: 4 mL, venous blood: 26 mL). GPS III system (Zimmer Biomet, Warsaw, IN, USA) (Figure 1A) was then used to collect 3 mL of LR-PRP from the anticoagulated blood. From this, 1 mL was used for component analysis and 2 mL was used for administration. The LR-PRP components are shown in Table 2. Under local anesthesia and fluoroscopy, 2 mL of PRP was intradiscally administered with a discography needle (Figure 1B). On the day of administration, the patient was hospitalized at a medical institution, and follow-up was arranged before being discharged the next day. 

### 2.3. Outcome Assessment

Primary endpoints were physical condition, laboratory data, and X-ray for safety evaluation. Secondary endpoints were MRI findings and pain scores using the visual analog scale (VAS), the Oswestry Disability Index (ODI), and the Roland–Morris Disability Questionnaire (RDQ) to evaluate PRP efficacy.

### 2.4. MRI

A clinical 3T MR system (Vantage Centurian: Canon Medical Systems Corporation) was used to perform all measurements. MRI was performed at 6 months after the PRP administration to measure the degree of inflammation using fat suppressed T2-weighted fast spin-echo sequence with the Dixon technique (fat suppressed T2WI) and T2* values from a gradient-echo (GRE) sequence with the 6-point Dixon method to evaluate the PRP administration effect on Modic change. In addition, proton density fat fraction (PDFF) mapping from the GRE sequence and the same 6-point Dixon method was used to quantify the fat marrow. The entirety of the two vertebrae showing Modic change was set as the region of interest (Figure 2) when measuring T2* values and PDFF. A previous paper has shown that MRI quantification of PDFF in vertebral bone marrow is highly accurate, repeatable, and reproducible among readers, field strengths, and MRI platforms [12]. In each patient, fat suppressed T2WI was performed by following scan parameters: TR 4435 ms/TE 99 ms, echo train length 19, slice thickness 3 mm, 1 number of excitation (NEX), field of view (FOV) 300 × 300 mm, 384 × 224 matrix, and 768 × 768 reconstruction matrix. For the PDFF mapping, the GRE sequence with the 6-point Dixon method was scanned using the following parameters: TR 7.6 ms/TE 1.2, 2.2, 3.2, 4.2, 5.2, and 6.2 ms; flip angle 3°; slice thickness 4 mm; 1 NEX; FOV 350 × 350 mm; 30 slices; and 256 × 256 matrix.

## 3. Results

### 3.1. Case 1

A 69-year-old female presented with chronic LBP complaints that had lasted for 8 months. The patient had previously undergone posterior L3/4 decompression surgery for lumbar spinal canal stenosis. Although the stenosis symptoms improved, the patient continued to have back pain postoperatively. Painkillers were ineffective and discontinued. She reported a pain intensity of 7/10 on the VAS, her RDQ was 13, and she had an ODI score of 40%. Patient demographics were shown in Table 3. Her X-rays showed degenerative change in the lumbar spine and the MRI showed Modic type 1 at the L2-L3 level (Figure 3A). Her LBP was hypothesized to be caused by Modic type 1, and PRP was administered to the intervertebral disc at the L2-L3 level. There were no adverse events in her physical condition or in laboratory data after the injection. Lumbar X-rays showed no iatrogenic disc injury. She was allowed to take NSAIDs, acetaminophen, duloxetine, and pregabalin if the pain worsened up to 48 h before the pain evaluation, but she did not require any. Laboratory data are shown in Table 4. Comparing the MRI before and 24 weeks after the PRP administration, the fat suppressed T2WI high signal volume decreased from 19.2 mL to 14.5 mL (Figure 4A) and the T2* value for the L2 and L3 vertebrae decreased from 4.73 msec to 4.23 msec. In addition, PDFF for the L2 and L3 vertebrae increased from 47.4 to 52.2%. The VAS temporarily improved to 4/10, but it returned to pretreatment levels 24 weeks after the injection. The ODI score decreased to 26% at 1 and 4 weeks after the injection and was still 30% at the last observation, lower than pretreatment. The RDQ decreased to 7 at 2 weeks after the injection and remained better than before the injection at 10 at the last observation. (Figure 5). 

### 3.2. Case 2

A 57-year-old male presented with chronic LBP, which had continued for 2 years. Due to lumbar spinal canal stenosis, he underwent L3/4 and L4/5 decompression, and his lower extremity symptoms resolved, but he still had residual LBP. Painkillers had no effect on his LBP. MRI showed Modic type 1 degeneration at the L4-L5 level (Figure 3B). His VAS was 5/10, his RDQ was 6, and his ODI score was 24%. Patient demographics were shown in Table 3. His LBP was hypothesized to be caused by Modic type 1, and the PRP was administered to the L4-L5 intervertebral disc. No adverse events were observed after the PRP administration. Disc height remained unchanged on lumbar X-rays after the administration. He also did not take any painkillers during the observation period. Laboratory data are presented in Table 4. Follow-up MRI showed that the fat suppressed T2WI high signal volume decreased from 17.7 mL to 16.3 mL (Figure 4B), the T2* value for L4 and L5 vertebrae decreased from 3.85 msec to 3.77 msec, and PDFF for L4 and L5 vertebrae increased from 45.7 to 52.0%. At 1 week after the injection, the VAS rose to 6/10, but then gradually declined, eventually improving to 2/10. The ODI score increased to 40% 1 week after the injection, but eventually improved to 18%. The RDQ tended to increase after the injection, but at 24 weeks after the injection, it decreased to four, showing an improvement from before the injection (Figure 5).

## 4. Discussion

Modic et al. reported on the MRI scans’ usefulness for spinal cord disease in 1983 [13], identified intensity changes in the endplates and subchondral bone on MRI as Modic change, and classified them into three types in 1988 [5,6]. Among the three types, Modic type 1 showed low signal intensity on T1WI and high signal intensity on T2WI, indicating edema and hypervascularity in the vertebral body [5]. Kusima et al. reported a significant correlation between Modic type 1 and LBP among the three types [7]. Florence et al. performed lumbar spine MRI on 2449 volunteers and reported that 5.8% had Modic change, which increases with age [14]. A recent systematic review and meta-analysis revealed a significantly higher prevalence of LBP in patients with Modic changes. However, Modic change was not associated with LBP severity or patient disability [15]. Modic type 1 findings are suggestive of inflammation and may be mixed with cases of infection. Ohtori et al. collected patients with Modic type 1, followed them for 2 years, and found that 4.2% developed pyogenic spondylitis [16]. In addition, Ninomiya et al. reported Modic type 1 has a risk of developing pyogenic discitis after posterior lumbar decompression surgery for lumbar spinal canal stenosis [17]. The Modic type 1 treatment is still controversial with some reports showing that antimicrobial agents are effective [18], whereas others reported that they were ineffective [19]. Sairyo et al. reported a patient with Modic type 1 who was successfully treated with endoscopic surgery, flushing, and drainage [20]. As Modic type 1 has various etiological factors, it may be necessary to develop a treatment plan for each case. More cases are expected to be reported in the future to establish the treatment for Modic type 1.

PRP therapy is one of the regenerative medicines that have been attracting attention in recent years. Platelets are produced by megakaryocytes as anucleated cells. A variety of growth factors, coagulation factors, adhesion molecules, cytokines, chemokines, and integrins are stored in platelets. PRP contains highly concentrated platelets and a number of plasma proteins associated with platelets during its preparation by centrifugation. PRP contains numerous growth factors that stimulate cellular anabolism, inflammatory mediators, and modulators that exert anti-inflammatory effects with fibrinogen acting as a biomaterial scaffold [9,21]. These actions are expected to have regenerative effects on tissue. PRP has been widely used in the clinical setting for tissue regeneration and repair [6,7]. Especially in the field of orthopedics, PRP has recently demonstrated regenerative abilities to repair injured tissues, including tendons, ligaments, and cartilage, all of which have a low intrinsic healing potential [8,9]. Matteo et al. reported PRP’s utility as a treatment option for refractory patellar and Achilles tendon disorders [11]. Kato et al. reported that PRP injection is effective, not only for ulnar collateral ligament partial tears, but also for complete tears, thus avoiding surgery [10]. Ye et al. reported that after an injection of PRP into the hip joint, hip osteoarthritis (OA) patients significantly improved their pain scores compared to intraarticular hyaluronic acid (HA) injection at 2 months after injection [22]. Zhang et al. reported PRP intraarticular injections reduced pain more effectively than HA injections in knee OA patients at 6 and 12 months after injection [23]. Furthermore, there are several reports of PRP administered into the intervertebral discs in patients with LBP, but the effects are varied [24,25,26,27,28,29]. Some of these studies diagnosed discogenic pain with discography, but given the multifactorial nature of LBP, it is possible that these patients’ LBP included in these previous reports may not be based on intervertebral disc degeneration. Systematic reviews and a meta-analysis showed that a PRP injection for lumbar disc disease was effective in controlling pain, but no structural improvement was found [30]. Meanwhile, there is no clinical PRP injection trial for the intervertebral disc of LBP patients with Modic type 1. Since the subjects in this study were limited to Modic type 1 LBP patients, the PRP’s effect on LBP could be evaluated in more detail compared to the previous reports. This study presented two patients in which there were no adverse events with PRP administration, and the pain scores using the VAS, ODI, and RDQ improved. Furthermore, MRI after PRP administration showed a decrease in the high signal volume of fat suppressed T2WI and T2* values with an increase in PDFF. These results suggest that the inflammation might have improved and have been replaced by fatty bone marrow through the PRP administration. In addition, the improvement in inflammation after PRP administration might be successfully quantified, which was the most important purpose of this study. Considering that fat suppressed T2WI signal quantification for Modic change has been reported to be highly reproducible [31], these results are considered reliable. Although there have been scattered reports in past studies of PRP administered to the intervertebral discs showing improvement in pain, few have demonstrated efficacy in MRI or other imaging evaluations. In this report, PRP efficacy was successfully confirmed using objective MRI as well as subjective patient-oriented outcomes. This study’s results suggest that a PRP intradiscal injection may have suppressed inflammation, thus improving pain. However, this report only presented two cases, and further case analysis is needed. Although the MRI was performed 6 months after administration in this study, we would like to evaluate over shorter time periods in the future as Modic change is considered to be dynamic in nature. The main limitation of this study is that it is difficult to determine whether Modic type 1 is the cause of LBP. Postoperatively, the patients’ lower extremity symptoms improved but LBP remained, and an MRI showed Modic type 1, which was diagnosed as being related to LBP. We believe this limitation can be overcome by increasing the number of cases. Therefore, this analysis is continued as a prospective study to establish PRP injection efficacy for Modic type 1 in the future.

## 5. Conclusions

This study’s results suggest that PRP administration into the intervertebral disc might improve LBP due to inflammation; PRP injection into the intervertebral disc of LBP patients with Modic type 1 might be safe and effective.

## Figures and Tables

**Figure 1 medicina-59-00112-f001:**
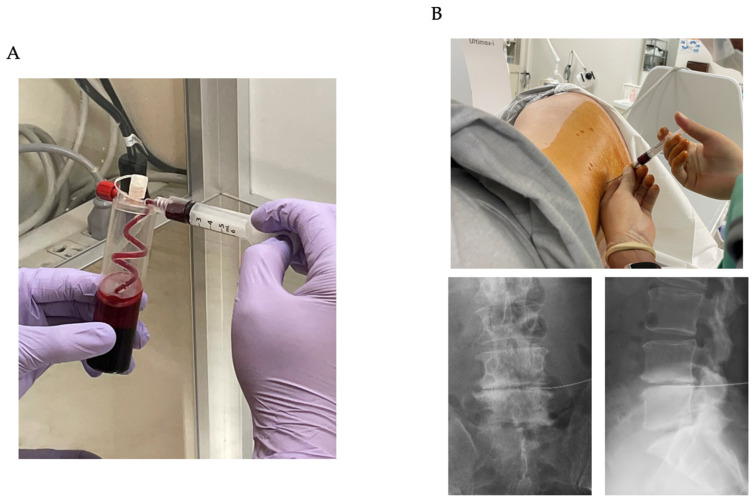
PRP preparation and procedure. (**A**) Using the GPS III system (Zimmer Biomet), 3 mL of PRP was collected from anticoagulated blood after centrifugation. (**B**) Under local anesthesia and fluoroscopy, 2 mL of PRP was administered into the disc with a discography needle.

**Figure 2 medicina-59-00112-f002:**
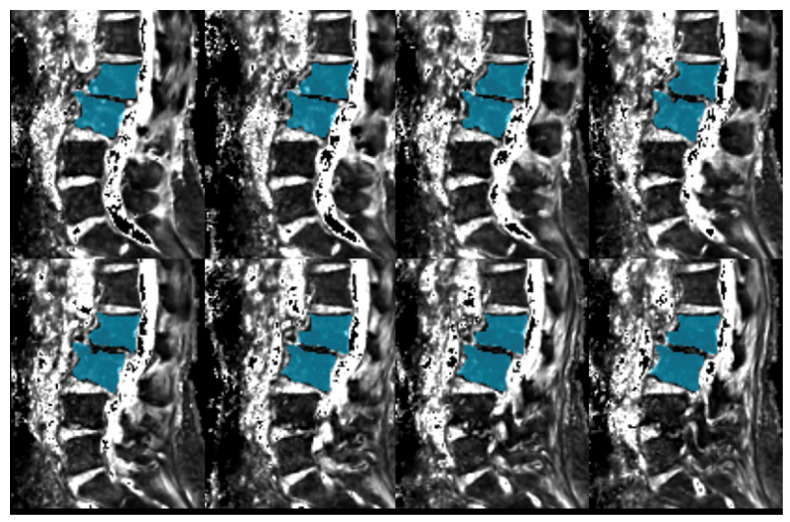
The region of interest for measuring T2* values and PDFF. The entirety of the two vertebrae showing Modic changes was set as the region of interest (blue color regions) when measuring T2* values and PDFF.

**Figure 3 medicina-59-00112-f003:**
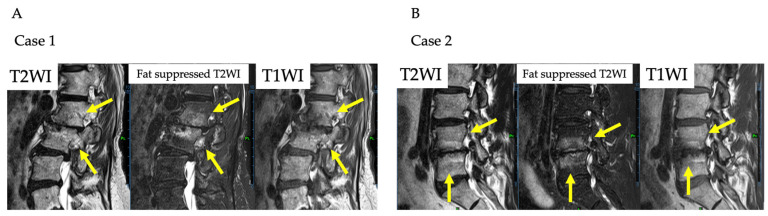
MRI images before PRP administration for case 1 and case 2. (**A**,**B**) are MRI images before PRP administration for case 1 and case 2, respectively, showing Modic type 1 at L2/3 in case 1 and at L4/5 in case 2 (yellow arrows).

**Figure 4 medicina-59-00112-f004:**
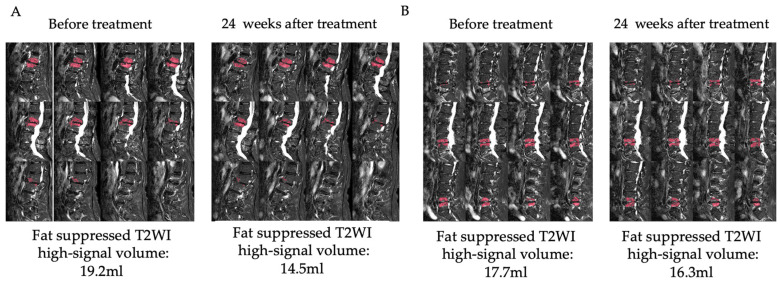
Images of fat suppressed T2WI before and 24 weeks after PRP administration in case 1 and case 2. (**A**,**B**) are images of fat suppressed T2WI before and 24 weeks after PRP administration in case 1 and case 2, respectively; fat suppressed T2WI high volume (red color regions) decreased from 19.2 mL to 14.5 mL in case 1 and from 17.7 mL to 16.3 mL in case 2.

**Figure 5 medicina-59-00112-f005:**
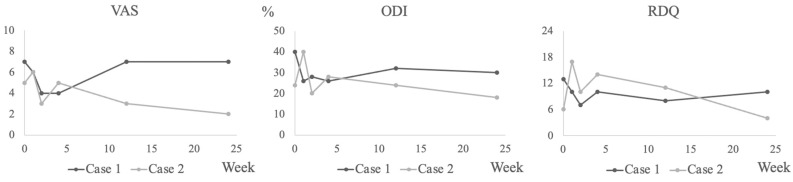
Scores of the VAS, ODI, and RDQ in case 1 and case 2. The VAS, ODI, and RDQ for case 1 and case 2 are shown. After PRP administration, the VAS in case 1 improved temporarily but returned to the same level as before administration at the last observation. All other scores in case 1 improved at the last observation compared to the pre-administration level. All scores in case 2 worsened temporarily after PRP administration but then gradually improved and eventually became lower than before administration.

**Table 1 medicina-59-00112-t001:** The inclusion and exclusion criteria.

Inclusion Criteria	(1) has participated in the informed consent process and is willing and able to sign an informed consent (2) aged from 20 to 70 years old (3) VAS pain intensity 50 mm or more for more than 3 months due to low back pain caused by intervertebral disc degeneration (4) intervertebral disc degeneration with Modic type 1 change (5) willing and able to complete scheduled follow-up evaluations as described in the study protocol
Exclusion Criteria	(1) BMI (body mass index): 30 or more (2) has a blood dyscrasia (platelet (Plt) less than 50,000/μL) (3) using anticoagulant or antiplatelet drug (4) has an autoimmune disease (5) has an active systemic inflammatory disease or infection (6) has a polyarthralgia (7) has another spinal disease, including vertebral fracture (8) has a compromised host status (diabetic, immune deficiency, chronic renal failure, hepatic cirrhosis, using immunosuppressive drug, etc.) (9) under treatment for malignant tumor (10) is known to be pregnant (11) judged as an inappropriate subject by surgeons performing regenerative medicine

**Table 2 medicina-59-00112-t002:** LR-PRP components.

	Case 1	Case 2
WBC (/μL)	12,900	17,800
Plt (10^4^/μL)	50.8	94.4
RBC (10^6^/μL)	1.62	1.58

**Table 3 medicina-59-00112-t003:** Patient demographics.

Variable	Case 1	Case 2
age	69 years	57 years
sex	female	male
height	1.61 m	1.67 m
weight	51 kg	73 kg
VAS	7/10	5/10
ODI	40%	24%
RDQ	13	6
pain duration	8 months	2 years

**Table 4 medicina-59-00112-t004:** Laboratory data.

Case 1	Week	Before Treatment	1	2	4	12	24
laboratory data	WBC (/μL)	4600	4500	4700	4300	4200	4700
	Plt (10^4^/μL)	15.8	-	16.2	16.7	15.6	16.3
	CRP (mg/dL)	0.03	0.07	0.08	0.03	0.07	0.03
	BUN (mg/dL)	18.2	14.7	15.5	21	17.2	16.4
	Cre (mg/dL)	0.57	0.55	0.57	0.62	0.58	0.59
Case 2	Week	Before Treatment	1	2	4	12	24
laboratory data	WBC (/μL)	4200	4000	4400	3800	4200	5500
	Plt (10^4^/μL)	23.1	20.8	21.5	22.8	21.1	23.5
	CRP (mg/dL)	0.01	0.04	0.02	0.02	0.02	0.03
	BUN (mg/dL)	18.3	17.5	16	16.2	14.6	18.4
	Cre (mg/dL)	0.89	-	-	0.8	0.82	0.86

## Data Availability

The datasets generated and/or analyzed during the current study are not publicly available due to limitations of ethical approval involving the patient data and anonymity but are available from the corresponding author on reasonable request.

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
