# Peer review of "Autologous Platelet-Rich Plasma Administration on the Intervertebral Disc in Low Back Pain Patients with Modic Type 1 Change: Report of Two Cases"

_medicina, 2023, doi:10.3390/medicina59010112_

Round 1

Reviewer 1 Report

Good proposition of PRP use.

Please evaluate change the "for" for "on" on title and whenever the expression "administration for the intervertebral disc" appears.

Abstract, line 18: This condition doesn't have a treatment or the current options are not efficient?

Materials and methods, line 61: There was more than one administration?

PRP preparation and procedure:

Line 65: Include details of the processing.

The PRP components were quantified? Quantification of platelets and leukocytes, at least, to classify it as a PRP rich or poor in WBC.

Line 69: Change to the past tense "will be".

Outcome assessment: Refine the description of the outcome assessment.

Figures: Use narrows or other visual indicative to point out the important features on images. Improve the description of the legends.

Results: Was observed any (statistically) significant difference on the measurements? 

The patients underwent to any other treatment after the PRP injection during the 6 months of evaluation? What were the recommendations after the injections?

Author Response

We sincerely thank the reviewer for reviewing our manuscript. Please find enclosed our revised manuscript and our point-by-point responses to the comment. The comment was very helpful to us when revising the manuscript. Changes made in the revised manuscript are indicated at track changes option. We feel that the manuscript has been significantly improved.

Reviewer 2 Report

The authors explored the safety and efficacy of platelet rich plasma (PRP) injection for two patients with low back pain (LBP) and Modic type 1 changes (MC1). Even though the study analyzed two patients, the study has novelty and it is interesting as there is no certain treatment for patients with LBP and MC1. However, study raises some comments. 

Major comments:

How can the patient be diagnosed to have LBP due to MC1?

The study lacks reporting the limitations of the study. Both patients are postoperative patients. What were the levels that have been operated? Could this be affected the results? Have the patients received any other treatments? 

Minor comments:

Why is the x-ray the primary endpoint? There is no information about x-rays in the results.

Should MRI be an endpoint if the authors are evaluating signal intensities?

Why is MRI evaluated 6 months after the injection? As MC are thought to be dynamic in nature, what would have been the situation for example 1 month after the injection?

As MC have various etiological factors, would certain MC1 cases be prone to better outcomes of the treatment? It seems that case 1 could have mechanical factors behind MC1 on MRI, for example. 

Abstract: I would suggest adding to conclusions that future studies are required as one can't draw any conclusions from two patient cases. 

Introduction: Are there any more recent articles/reviews regarding PRP treatments?

Discussion: I would suggest to cite more comprehensive/relevant articles/reviews regarding MC to be more comprehensive. 

The authors state certain PRP studies and their outcomes. Are there any reviews/meta-analyses to show regarding the efficacy of PRP injections? 

The authors state that "These results suggested that the inflammation improved and has been replaced by fatty bone marrow by PRP administration". Are the changes in signal intensities significant? Is the inflammation truly improved? 

Abbreviations should be consistently used. 

Round 2

Reviewer 2 Report

The authors have revised the manuscript and responded to the comments nicely. I have only one comment.

Due to challenges to "diagnose" LBP patient to have pain due to MC1, I would suggest to modify and soften the following sentences: 

case 1: Patient was diagnosed with LBP due to Modic type 1

case 2: Diagnosed with LBP due to Modic type 1

LBP can only be hypothesized to be caused by MC1 as this cannot be confirmed.
